# Effects of a Novel Dispersible Supplement Containing 2500 IU of Vitamin D and 1000 µg of B12 in Restoring Vitamin D and B12 Insufficiency: A Multicenter, Randomized Controlled Trial

**DOI:** 10.3390/nu17030419

**Published:** 2025-01-23

**Authors:** Nikolaos Angelopoulos, Rodis D. Paparodis, Ioannis Androulakis, Anastasios Boniakos, Sarantis Livadas

**Affiliations:** 1Hellenic Endocrine Network, Ermou 6 Str., 10563 Athens, Greece; rodis@paparodis.gr (R.D.P.); iandroulakis@gmail.com (I.A.); anbonendo74@gmail.com (A.B.); sarntis@gmail.com (S.L.); 2Private Practice, Endocrinology, Diabetes and Metabolism Clinics, Venizelou Str., 65302 Kavala, Greece; 3Division of Endocrinology, Diabetes and Metabolism, Loyola University Medical Center, Maywood, IL 60153, USA; 4Edward Hines Jr. VA Hospital, Hines, IL 60141, USA; 5Private Practice, Endocrinology, Diabetes and Metabolism Clinics, Gerokostopoulou 24, 26221 Patra, Greece; 6Private Practice, Endocrinology, Diabetes and Metabolism Clinics, Tzanaki Emmanouil 17, 73134 Chania, Greece; 7Private Practice, Endocrinology, Diabetes and Metabolism Clinics, Omirou 3, 13231 Athens, Greece; 8Endocrine Unit, Athens Medical Centre, 15125 Athens, Greece

**Keywords:** vitamin D, vitamin B12, vitamin insufficiency, supplement

## Abstract

Background/Objectives: Vitamins D and B12 play a crucial role in maintaining bone health, immune function, and neurological integrity. Combined deficiencies in these vitamins can lead to severe health consequences. Current treatment approaches, such as dietary changes and single-vitamin supplementation, often fail to address these deficiencies comprehensively. This study evaluates the effectiveness of concurrent vitamin D and B12 supplementation to correct these insufficiencies. Methods: A prospective, multicenter, randomized controlled trial was conducted in Greece from October 2024 to December 2024. Participants aged 20 to 80 years, with insufficient levels of 25-hydroxyvitamin D (serum < 20 ng/mL) and B12 (serum < 250 ng/L), were eligible for inclusion. Results: A total of 124 patients were randomized into three groups: one receiving vitamins B12 and D in a single supplement (2500 IU + 1000 mcg), one receiving separate doses of each vitamin (2000 IU + 1000 mcg), and a control group receiving no supplementation. The results demonstrated a significant increase in B12 and 25-hydroxyvitamin D levels among the supplemented groups. Particularly, participants in the combined supplementation group showed higher average serum levels of both vitamins. By the end of this study, 37.1% of those in the combined supplement group achieved adequate vitamin levels, compared to 29.4% in the separate supplementation group. Conclusions: In conclusion, combined supplementation may improve patient adherence and compliance, leading to better health outcomes for individuals with combined vitamins D and B12 deficiencies.

## 1. Introduction

Vitamins D and B12 are vital nutrients essential for maintaining numerous physiological functions, including bone health [1], immune responses [2], and neurological integrity. Vitamin D plays a critical role in calcium and phosphorus absorption, which are fundamental for maintaining healthy bone density and structure. Adequate levels of vitamin D help prevent conditions such as osteoporosis and rickets. Additionally, vitamin D has immunomodulatory effects, enhancing the body’s immune response by promoting the activity of immune cells such as macrophages and T cells, thereby offering protection against infections and autoimmune diseases. On the other hand, vitamin B12 is indispensable for neurological health and function. It is a key player in the synthesis of myelin, the protective sheath around nerve fibers, and in the production of neurotransmitters, which are essential for effective communication between nerve cells [3]. Vitamin B12 also supports cognitive functions and has been linked to a reduced risk of neurodegenerative diseases [4]. Moreover, it is crucial for red blood cell formation and DNA synthesis, processes that are vital for overall cellular function and energy production [5]. Deficiencies in either of these vitamins can lead to severe health consequences, underscoring their importance in maintaining overall health and well-being.

Despite their importance, deficiencies in these vitamins are widely prevalent worldwide, affecting various demographic groups with significant health consequences. Vitamin D deficiency has been linked to an increased risk of osteoporosis, cardiovascular diseases [1], and certain cancers [5]. At the same time, a lack of vitamin B12 can result in anemia, neurological disorders, and cognitive decline [6].

Current interventions, such as dietary modifications and single-vitamin supplementation, often fail to address these deficiencies comprehensively [7]. Supplementation with single vitamins often faces complexities such as variable absorption rates, differences in individual metabolic responses, and the need for ongoing compliance [8]. Additionally, certain populations, including the elderly and those with gastrointestinal disorders, may not adequately absorb oral supplements, necessitating alternative forms of administration [9]. This study explores the efficacy of a combined vitamin D and B12 supplement, hypothesizing that this approach might offer synergistic benefits and greater convenience. By evaluating the supplement’s impact on serum vitamin levels and associated health outcomes, this work aims to improve clinical practice and public health strategies for addressing these widespread nutritional deficiencies in a more effective way.

## 2. Materials and Methods

### 2.1. Study Protocol

The present study is a prospective, multicentric, open-label, randomized controlled trial conducted at 5 outpatient endocrine clinical centers in Greece, according to a controlled, randomized, and repeated measures design from October 2024 to December 2024 (registration code: ACTRN12624000468527).

Patients were eligible to participate in the present study if they were >20 and <80 years of age, had vitamin D concentration lower than 20 ng/mL, and vitamin B12 concentration lower than 250 ng/L.

Exclusion criteria were as follows: 1. Individuals taking vitamin D supplements within 6 months before the study initiation. 2. The use of medications known to interfere with vitamin D metabolism or absorption, such as antiepileptic drugs (phenytoin, phenobarbital), glucocorticoids (prednisone), or certain weight-loss medications (orlistat). 3. Chronic kidney disease. 4. Other metabolic disorders: individuals with specific metabolic disorders that could impact vitamin D metabolism or lead to altered vitamin D levels, such as primary hyperparathyroidism, hyperthyroidism, or malabsorption syndromes (e.g., celiac disease, Crohn’s disease). 5. Conditions that significantly affect bone health, such as osteoporosis, osteomalacia, or Paget’s disease. 6. Pregnancy and breastfeeding. 7. Severe illnesses or comorbidities: terminal cancer, uncontrolled diabetes, or autoimmune disorders. 8. Drugs that affect vitamin b12 metabolism: Nitrous oxide, Proton pump inhibitors, H2 receptor antagonists, Metformin, Colchicine, Slow K (potassium chloride) preparations, and Cholestyramine. 9. History of bariatric surgery or other gastric operation. 10. Active alcohol abuse. 11. During this study, patients who received vitamin D or B12 containing supplements outside of the study protocol.

At visit 1 (study initiation), patients were randomly allocated in one of 3 groups. A neurochemical compound (Fortius D3 + B12^®^ tablets, 2500 IU of vitamin D and 1000 mcg of vitamin B12, Geoplan Nutraceuticals, 90 tablets, one dispersible tablet daily, independently of food intake) was prescribed to patients comprising Group A. Group B consisted of patients who were advised to take two separate tablets: vitamin D 2000 IU (D3-fix^®^, Unipharma Pharmaceuticals, Athens, Greece) administrated after lunch and vitamin B12 (B12-fix^®^ 1000 mcg, Unipharma Pharmaceuticals, Athens, Greece) after breakfast. The randomization ratio was 3:3:1 for Groups A, B, and C, respectively. Compliance with therapy intake was checked via a questionnaire during the final visit (a self-reported questionnaire that assesses adherence to medication refills and taking behavior). The control group (Group C) consisted of patients who did not receive any supplements during the study period. After 12 weeks of treatment, patients were reassessed (visit 2), and this study ended. The randomization sequence was generated using a computer-based random number generator. Both participants and investigators were aware of the randomization process in this study, as it was not conducted under blinded conditions.

### 2.2. Measurements

During the study period, blood samples were taken twice: at the beginning of this study (visit 1) and at the end of this study (follow-up visit 2) to assess the serum levels of B12 and vitamin D. Moreover, in visit 1, height and weight were measured, and body mass index (BMI) was calculated as the ratio of weight (in kilograms) to squared height (in meters). Both serum B12 and vitamin D were estimated by the chemiluminescent microparticle immunoassay (CMIA) technology (Abbott Diagnostics, Abbott Park, IL, USA).

### 2.3. Study Setting

This study was conducted in five private Endocrinology, Diabetes and Metabolism clinics scattered throughout Greece. Data were transferred from all participating sites to the principal investigator (N.A) for assessment and analysis. Informed consent was obtained from all participants before their inclusion in this study. The consent process included a clear explanation of this study’s purpose, procedures, potential risks, and benefits, and participants had the opportunity to ask questions. Furthermore, participant confidentiality was strictly maintained throughout this study.

### 2.4. Ethical Approval

This study was conducted in accordance with the principles of the Declaration of Helsinki and was approved by the Institutional Review Board of the Hellenic Endocrine Network, Athens, Greece (IRB approval number: Ν2024/0121311, 18 January 2024).

### 2.5. Statistical Analysis

The normality of the distribution of our data was analyzed using the Kolmogorov–Smirnov test. Differences in data with skewed distributions were analyzed with the Wilcoxon nonparametric test. The Kruskal–Wallis test was used to examine significant differences between groups concerning dependent variables. A Dunn–Bonferroni test was then used to compare the groups in pairs to investigate which was significantly different. The results were presented as the median and interquartile range (IQR) for normally distributed data and as mean values ± standard deviation (SD) for abnormally distributed parameters. Moreover, the difference between time points was indicated in percentage. Statistical analyses were performed using MedCalc for Windows, version 19.4 (MedCalc Software, Ostend, Belgium), and *p*-values < 0.05 were deemed significant.

### 2.6. Funding

The present study is an investigator-initiated research protocol, funded entirely by a research grant provided by Geoplan Neutraceuticals, Athens, Greece. The sponsor did not participate in this study conception or design, the randomization process, the study procedures, the data collection, the statistical analysis, or the writing of the present manuscript.

## 3. Results

One hundred and forty-four subjects who met our eligibility criteria were initially enrolled in this study and randomized to receive a compound of B12 + D3 supplement (Group A, n = 62), separated supplements (Group B, n = 62), and the remainder received no supplement (control group C, n = 20). Of those, eleven patients in Group B reported low treatment compliance at visit 2 (defined as taking less than 80% of the recommended doses) and were thus excluded from this study. Four patients in the control group were also excluded since they started various supplements containing either vitamin B12 and/or vitamin D (Figure 1).

No serious adverse events occurred during this study. The remaining 129 patients who completed this study were included in the present analysis (visit 2).

The mean age of the overall population was 45.68 ± 13.79 years (range of 20–79 years, 19 males/110 females). Table 1 and Figure 2 show the main characteristics of the study population at the baseline. There were no differences among the three groups in the baseline demographics and laboratory findings except for the levels of B12, which were slightly higher in the control group (229.13 ± 17.72, *p* < 0.01).

There was a significant increase in both B12 and 25(OH)D3 levels in the treated groups (A and B). In Group A, B12 increased from 200.63 ± 31.01 to 438.2 ± 130.1, (*p* < 0.01), and 25(OH)D3 increased from 18.39 ± 3.61 to 30.2 ± 5.4, (*p* < 0.01). In Group B, B12 increased from 198.92 ± 29.33 to 374.5 ± 78.6, (*p* < 0.01), and 25(OH)D3 from 18.67 ± 3.97 to 29 ± 5.7 (*p* < 0.01), as illustrated in Figure 3A,B.

The levels of B12 in Group A were significantly higher than those in Group B after treatment, *p* = 0.017. Comparisons of the vitamin level changes with the therapy between the three groups are illustrated in Table 2.

After treatment, B12 increased by 120.5% on average in Group A, significantly more compared to the increase (90.5%) detected in Group B (*p* = 0.015). The percentage increase in 25(OH)d3 levels was also greater in Group A (69.1%) than in Group B (59.5%) although the difference showed borderline statistical significance (*p* = 0.052) (Table 3).

In terms of achieving adequate vitamin levels with therapy (B12 > 350 mg/dl and vitamin D > 30 ng/mL), we found that 23 out of 62 patients who received the compound supplement (37.1%) achieved that threshold. A total of 43 (69.4%) restored adequate levels of B12, and 33 (53.2%) had adequate vitamin D levels. In addition, 23 of 51 patients on separate supplements achieved adequate vitamin D levels (45.1%) and 30 (58.8%) adequate levels of b12, while only 15 (29.4%) had adequate levels of both b12 and vitamin D at the end of this study (differences did not reach statistical significance).

## 4. Discussion

Despite efforts to increase the intake of vitamin D and B12 through dietary modifications, many individuals are unable to achieve adequate concentrations due to factors such as limited dietary sources, poor absorption, and dietary restrictions, particularly in populations with specific dietary preferences or medical conditions.

Natural food sources of vitamin D are few, such as fatty fish (e.g., salmon, mackerel), fortified foods (e.g., milk, orange juice), and egg yolks, but these may not provide sufficient amounts for many people [10].

Conditions like obesity [10], chronic kidney disease [11], or malabsorption syndromes (e.g., celiac disease, Crohn’s disease) [12] impair vitamin D absorption and metabolism.

Due to the variability in sun exposure, especially during the winter months, many people may need to rely on dietary sources of vitamin D or supplements to maintain optimal levels [13]. To avoid this bias, our study was conducted during winter for 3 months from October to December, when vitamin D concentrations are expected to decrease [14].

Vegetarians and vegans are at higher risk for vitamin B12 deficiency [15] because plant-based foods naturally lack this vitamin. B12 is mainly found in animal products like meat, fish, dairy, and eggs. Older adults and individuals with gastrointestinal conditions (e.g., pernicious anemia, gastritis, or after bariatric surgery) often have reduced intrinsic factors or gastric acid secretion, which are required for vitamin B12 absorption [16,17]. Furthermore, the long-term use of medications such as metformin [18] or proton pump inhibitors (PPIs) can interfere with vitamin B12 absorption [19].

The rising incidence of concomitant vitamin B12 and vitamin D deficiencies, particularly in vulnerable populations such as the elderly, those with chronic diseases, and individuals with restrictive diets, underscores the urgent need for targeted treatment strategies [20]. Proactive interventions, including routine screening, supplementation, and public health initiatives, are critical to mitigate the long-term health risks of these deficiencies. By addressing these nutrient gaps early, we can improve quality of life, prevent complications, and reduce the overall healthcare burden, particularly in developing countries where access to these nutrients is often limited.

A systematic review [21] confirmed that the response to supplementation is dose-dependent, with higher doses resulting in predictable increases in 25(OH)D levels. A human study investigated the effects of daily supplementation with 2000 IU of vitamin D₃ over 16 weeks [22]. The study found that this dosage effectively raised serum 25(OH)D levels to above 75 nmol/L (30 ng/mL) in most participants, achieving vitamin D sufficiency. Another study, as reported in *Life*, compared the effects of daily doses of 1000 IU and 2000 IU of vitamin D₃ in healthy individuals [23]. The results indicated that taking 2000 IU daily led to a significant increase in serum 25(OH)D levels, which plateaued approximately 30 days after the discontinuation of supplementation.

A systematic review and network meta-analysis concluded that oral administration of vitamin B12 is as effective as intramuscular (IM) and sublingual (SL) routes in elevating serum B12 levels, with no significant differences among these methods [24]. The American Academy of Family Physicians (AAFP) suggests an oral replacement dosage of 1000 to 2000 mcg per day for one to two weeks, followed by a maintenance dose of 1000 mcg daily for life [25].

To our knowledge, no studies directly address the efficacy of combined vitamin D and B12 supplementation. As such, this research fills an important gap by exploring the effectiveness of this strategy in restoring both vitamin adequacy. A single tablet is more convenient than taking multiple pills. This reduces the complexity of the supplementation regimen, rendering it easier for individuals to remember and adhere to it, particularly when multiple medical conditions, such as hypertension [26] and diabetes [27] are present. Individuals might perceive a combination supplement as more efficient and beneficial, which can positively influence their commitment to consistent use. In our small cohort, 18% of patients allocated to separate pills (group B) exhibited poor compliance with their treatment affecting the efficacy of the prescribed supplements in restoring vitamin adequacy. By addressing these factors, a combination supplement can effectively improve adherence and compliance, leading to better health outcomes for individuals in need of supplementation for vitamins D and B12.

## 5. Conclusions

One limitation of our study is the relatively short duration of the research period, which may not fully capture long-term trends or outcomes. Another limitation is the narrow demographic profile of the participants, which may limit the generalizability of the findings to broader populations. Given the limited number of relevant studies, additional research is necessary to validate and extend our findings. Longitudinal studies with larger cohorts may provide further evidence and facilitate the investigation of the efficacy of combined supplementation in populations with diverse dietary patterns or underlying health conditions. Furthermore, a comprehensive assessment of potential adverse effects or drug interactions over prolonged periods is warranted.

## Figures and Tables

**Figure 1 nutrients-17-00419-f001:**
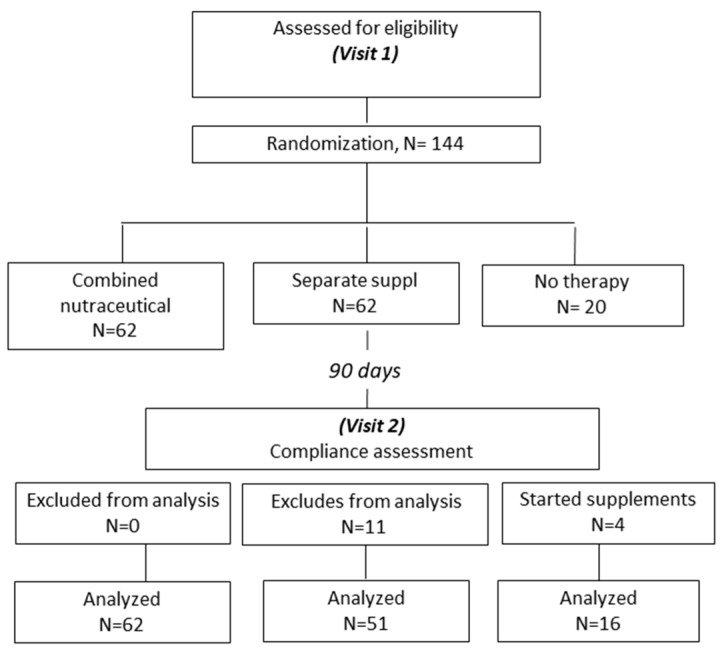
Study chart flow.

**Figure 2 nutrients-17-00419-f002:**
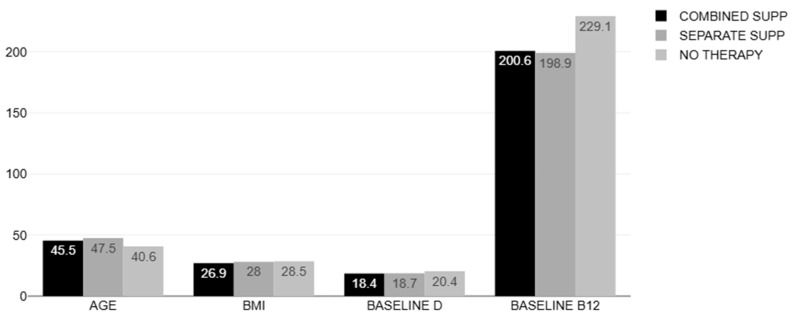
Characteristics of the study population.

**Figure 3 nutrients-17-00419-f003:**
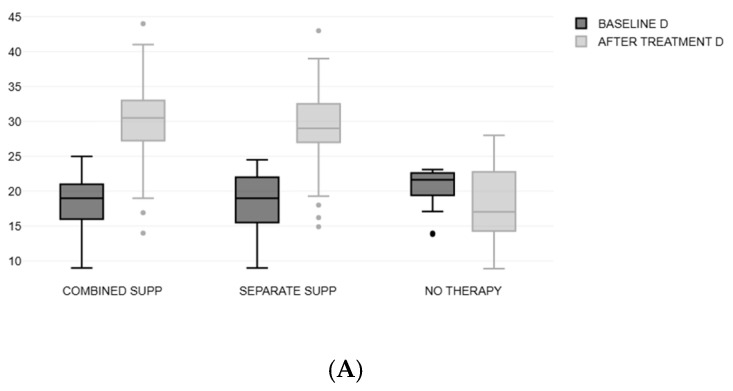
(**A**,**B**) Changes in vitamin D levels and B12 levels.

**Table 1 nutrients-17-00419-t001:** Baseline characteristics of the study population.

Variable	Total *n* = 129	Group A *n* = 62	Group B *n* = 51	Control *n* = 16	*p*
Age	45.68 ± 13.79	45.49 ± 14.78	47.49 ± 12.73	40.62 ± 12.5	ns
Gender female, *n* (%)	119 (85.27)	52 (83.87)	44 (86.27)	14 (87.50)	ns *
BMI (kg/m^2^)	27.53 ± 4.89	26.86 ± 5.41	24.76 ± 3.53	28.47 ± 5.48	ns
Β12 (mg/dL)	203.49 ± 30.42	200.63 ± 31.01	198.92 ± 29.33	229.13 ± 17.72	0.001
25(OH)D (ng/mL)	18.75 ± 3.73	18.39 ± 3.61	18.67 ± 3.97	20.41 ± 3.1	ns

Footnote: *p*: one-way analysis of variance Kruskal–Wallis-Test; *p* *: x^2^ test of independence. ns: not significant. All values are expressed as the mean ± standard deviation.

**Table 2 nutrients-17-00419-t002:** Biochemical profiles before and after treatment.

Variable	Group A *n* = 62			Group B *n* = 51			Control *n* = 16			
	Baseline	Study End	*p*	Baseline	Study End	*p*	Baseline	Study End	*p*	*p*
Β12 (mg/dL)	200.63 ± 31.01	438.2 ± 130.1	<0.001	198.92 ± 29.33	374.5 ± 78.6	<0.001	229.13 ± 17.72	252.4 ± 41.9	0.39	<0.001 ^a^
25(OH)D (ng/mL)	18.39 ± 3.61	30.2 ± 5.4	<0.001	18.67 ± 3.97	29 ± 5.7	<0.001	20.41 ± 3.1	18.2 ± 5.8	0.06	<0.001 ^b^

*p*: Nonparametric Wilcoxon test for paired values; *p*: Kruskal–Wallis test and Dunn’s multiple comparison test between groups (mean rank difference). ^a^ Group A vs. Group B: *p* = 0.017; Group A vs. Control: *p* < 0.001; Group B vs. Control: *p* < 0.001. ^b^ Group A vs. Group B: *p* = 0.293; Group A vs. Control: *p* < 0.001; Group B vs. Control: *p* < 0.001.

**Table 3 nutrients-17-00419-t003:** Vitamin level alterations in means and percentile changes after treatment.

	Difference				Percentage of Difference, %			
Variable	Β12 (mg/dL)		25(OH)D (ng/mL)		Β12 (mg/dL)		25(OH)D (ng/mL)	
	*p* = 0.007		*p* = 0.044		*p* = 0.015		*p* = 0.052	
	Group A	Group B	Group A	Group B	Group A	Group B	Group A	Group B
Mean	237.6	175.6	11.8	10.4	120.5	90.5	69.1	59.5
95% (CI of Mean)	206.8–268.4	154.8–196.3	10.5–13.2	9–11.8	104.3–136.8	78.8–102.1	59.1–79.1	49.4–69.7
SD	121.2	73.7	5.3	5	64	41.5	39.4	36
Min	67	0	−5	3.1	37.2	0	20.8	17.7
Max	661	390	22.9	23	361.9	229.9	222.2	176.9

*p*: Wilcoxon Test.

## Data Availability

The original contributions presented in this study are included in the article. Further inquiries can be directed to the corresponding author.

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
