# Peer review of "Effects of a Novel Dispersible Supplement Containing 2500 IU of Vitamin D and 1000 µg of B12 in Restoring Vitamin D and B12 Insufficiency: A Multicenter, Randomized Controlled Trial"

_nutrients, 2025, doi:10.3390/nu17030419_

Round 1

Reviewer 1 Report

Comments and Suggestions for Authors

This carefully described nutritional trial tested the efficiency of improving vitamin D status and vitamin B12 status by supplying the supplements as a combined tablet to be consumed daily or by supplying the two vitamins as separate tablets each to be consumed after different meals each day. The results indicated that the group of subjects taking the combined supplements had greater improvements of vitamin D stratus and vitamin B12 status than the group who were taking these supplements separately.

Apparently the rationale behind this study is explained at line 63: “…hypothesizing that this approach might offer synergistic benefits and greater convenience.”  It is not clear in the manuscript what the basis of this hypothesis might be. Was there evidence before undertaking this trial to substantiate the reasons for testing the hypothesis?

Again, in the Discussion, Line 219, it is stated that “a single tablet is more convenient than taking multiple tablets.” This is hardly a strong justification for undertaking this study.

Minor points:

Line 27 in the Abstract refers to “vitamin D levels”. Vitamin D concentration was not measured. The levels referred to were those of 25-hydroxyvitamin D. It is ambiguous to use the term “vitamin D levels” because both 25-hydroxyvitamin D and vitamin D are present in the blood serum or plasma samples being analysed.

In Tables 1 and 2 the units for the concentration of 25(OH)D in blood serum are defined as mg/dL. The values are not as high as that. The usual unit of concentration of 25(OH)D is ng/ml which if being expressed per 100 ml or 1 deciliter, would be micrograms per deciliter, not milligrams per deciliter.

Author Response

Comment 1:  

Apparently the rationale behind this study is explained at line 63: “…hypothesizing that this approach might offer synergistic benefits and greater convenience.”  It is not clear in the manuscript what the basis of this hypothesis might be. Was there evidence before undertaking this trial to substantiate the reasons for testing the hypothesis?Again, in the Discussion, Line 219, it is stated that “a single tablet is more convenient than taking multiple tablets.” This is hardly a strong justification for undertaking this study.

  Response:

You are correct that the rationale behind the hypothesis, as stated at line 63, is not substantiated in the manuscript. In everyday practice, the recommendation to take multiple dietary supplements to patients who are already taking various medications can be problematic. We thus shoot to determine if a combined supplement could result to a better adherence in those patients.

Minor points:

Comment 2:

Line 27 in the Abstract refers to “vitamin D levels”. Vitamin D concentration was not measured. The levels referred to were those of 25-hydroxyvitamin D. It is ambiguous to use the term “vitamin D levels” because both 25-hydroxyvitamin D and vitamin D are present in the blood serum or plasma samples being analysed.

Response:

Thank you for pointing out the ambiguity in our reference to "vitamin D levels" in the abstract. We acknowledge that 25-hydroxyvitamin D, rather than vitamin D, was measured and agree that it is important to use precise terminology to avoid confusion. We revised the text to specify "25-hydroxyvitamin D levels" to ensure clarity and accuracy (lines 23 and 27).

 Comment 3:

In Tables 1 and 2 the units for the concentration of 25(OH)D in blood serum are defined as mg/dL. The values are not as high as that. The usual unit of concentration of 25(OH)D is ng/ml which if being expressed per 100 ml or 1 deciliter, would be micrograms per deciliter, not milligrams per deciliter.

Response:

Thank you for catching the error in the units reported for 25(OH)D concentrations in Tables 1 and 2. It was a typo error. All values are referred to ng/ml and are now corrected throughout the manuscript.

Reviewer 2 Report

Comments and Suggestions for Authors

Review Summary

The article explores the effects of a novel dispersible supplement containing Vitamin D and B12, offering valuable insights into addressing dual deficiencies through a single combined supplement. The study design is robust, featuring a randomized controlled trial across multiple clinical centers, which strengthens its credibility. The results highlight significant improvements in vitamin levels and adherence, making this research a useful contribution to the field of endocrinology and public health.

Strengths:

Clear Objectives: The study addresses a relevant health concern and provides practical implications for improving patient compliance with supplementation.

Study Design: The randomized controlled trial across various clinics in Greece enhances the reliability of the results.

Comprehensive Analysis: The statistical methods used are appropriate, and the results are clearly presented with tables and figures.

Areas for Improvement:

Clarity in Methods:

The description of the randomization process could be more detailed. How was the randomization sequence generated, and was it concealed from the participants and investigators?

Discussion Depth:

The discussion section could benefit from a deeper comparison with previous studies. Are the results consistent with existing literature, and what unique contribution does this study make?

Ethical Considerations:

The ethical approval section mentions compliance with the Declaration of Helsinki, but it would be beneficial to elaborate on how informed consent was obtained and how participant confidentiality was ensured.

Limitations:

A clear limitations section would strengthen the manuscript. Consider discussing any potential biases, such as the limited duration of the study or the specific demographic of the participants.

Future Directions:

Adding a section on potential future research directions would be helpful. For instance, exploring the long-term effects of combined supplementation or its impact on different populations.

Recommendations for Future Research:

Investigate the long-term sustainability of the observed benefits in vitamin levels.

Explore the effectiveness of combined supplementation in populations with different dietary habits or health conditions.

Evaluate potential side effects or interactions with other medications over an extended period.

Author Response

Clarity in Methods:

Comment 1

The description of the randomization process could be more detailed. How was the randomization sequence generated, and was it concealed from the participants and investigators?

Response:

Thank you for highlighting the need for a more detailed description of the randomization process. The randomization sequence was generated using a computer-based random number generator. This study was not conducted in a blind-randomization manner and thus was not concealed from the participants and the investigators. We added this information to the Materials and Methods section (lines 103-105).

Discussion Depth:

 Comment 2:

The discussion section could benefit from a deeper comparison with previous studies. Are the results consistent with existing literature, and what unique contribution does this study make?

Response:

We agree with your comment. We enriched the discussion with the relevant literature (lines 225-242).

Ethical Considerations:

Comment 3:

The ethical approval section mentions compliance with the Declaration of Helsinki, but it would be beneficial to elaborate on how informed consent was obtained and how participant confidentiality was ensured.

Response:

We added this information to the relevant section (line 117).

Limitations:

Comment 4:

 A clear limitations section would strengthen the manuscript. Consider discussing any potential biases, such as the limited duration of the study or the specific demographic of the participants.

Response:

Thank you for your valuable feedback regarding the inclusion of a limitations section. In response, we included a dedicated limitations section highlighting factors such as the study's limited duration and the specific demographic of participants (lines 253-257).

Future Directions:

Comment 5:

Adding a section on potential future research directions would be helpful. For instance, exploring the long-term effects of combined supplementation or its impact on different populations.

Recommendations for Future Research:

Investigate the long-term sustainability of the observed benefits in vitamin levels. Explore the effectiveness of combined supplementation in populations with different dietary habits or health conditions. Evaluate potential side effects or interactions with other medications over an extended period.

Response:

We added a separate paragraph at the end of the discussion regarding recommendations and future research (lines 256-261).